# HIV-1 Envelope Glycoprotein Amino Acids Signatures Associated with Clade B Transmitted/Founder and Recent Viruses

**DOI:** 10.3390/v11111012

**Published:** 2019-11-01

**Authors:** Alexis Kafando, Christine Martineau, Mohamed El-Far, Eric Fournier, Florence Doualla-Bell, Bouchra Serhir, Adama Kazienga, Mohamed Ndongo Sangaré, Mohamed Sylla, Annie Chamberland, Hugues Charest, Cécile L. Tremblay

**Affiliations:** 1Département de microbiologie, infectiologie et immunologie, Faculté de médecine, Université de Montréal, Montréal, Québec QC H3T 1J4, Canada; alexis.kafando@umontreal.ca (A.K.); hugues.charest@inspq.qc.ca (H.C.); 2Laboratoire de santé publique du Québec, Institut national de santé publique du Québec, Sainte-Anne-de-Bellevue, Québec QC H9X 3R5, Canada; christine.martineau@canada.ca (C.M.); eric.fournier@inspq.qc.ca (E.F.); florence.doualla-bell@inspq.qc.ca (F.D.-B.); bouchra.serhir@inspq.qc.ca (B.S.); 3Centre de recherche du centre hospitalier de l’Université de Montréal, Montréal, Québec QC H3T 1J4, Canada; mohamed.el.far.chum@ssss.gouv.qc.ca (M.E.-F.); syllmoh@yahoo.fr (M.S.); chamberland.annie@videotron.ca (A.C.); 4Faculty of Science, Hasselt University, 3500 Hasselt, Belgium; kazienga_adama@yahoo.fr; 5Département de médecine sociale et préventive, École de santé publique, Université de Montréal, Montréal QC H3T 1J4, Québec, Canada; ndongosangare@yahoo.fr

**Keywords:** HIV-1, acute/early infection, transmitted/founder viruses, recent viruses, envelope, amino acids, genetic signatures, signal peptide, cytoplasmic domain, lentivirus lytic peptide segment 1

## Abstract

Background: HIV-1 transmitted/founder viruses (TF) are selected during the acute phase of infection from a multitude of virions present during transmission. They possess the capacity to establish infection and viral dissemination in a new host. Deciphering the discrete genetic determinant of infectivity in their envelope may provide clues for vaccine design. Methods: One hundred twenty-six clade B HIV-1 consensus envelope sequences from untreated acute and early infected individuals were compared to 105 sequences obtained from chronically infected individuals using next generation sequencing and molecular analyses. Results: We identified an envelope amino acid signature associated with TF viruses. They are more likely to have an isoleucine (I) in position 841 instead of an arginine (R). This mutation of R to I (R841I) in the gp41 cytoplasmic tail (gp41CT), specifically in lentivirus lytic peptides segment 1 (LLP-1), is significantly enriched compared to chronic viruses (OR = 0.2, 95% CI (0.09, 0.44), *p* = 0.00001). Conversely, a mutation of lysine (K) to isoleucine (I) located in position six (K6I) of the envelope signal peptide was selected by chronic viruses and compared to TF (OR = 3.26, 95% CI (1.76–6.02), *p* = 0.0001). Conclusions: The highly conserved gp41 CT_ LLP-1 domain plays a major role in virus replication in mediating intracellular traffic and Env incorporation into virions in interacting with encoded matrix protein. The presence of an isoleucine in gp41 in the TF viruses’ envelope may sustain its role in the successful establishment of infection during the acute stage.

## 1. Introduction

HIV-1 genetic diversity due to frequent mutation rates, polymorphisms, recombination events and altered pattern of glycosylation within the envelope (Env) [1] drives HIV-1 escape from broadly neutralizing antibodies (bnAbs) and other mechanisms developed by the immune system [2,3,4]. HIV-1 envelope polymorphism determines the functional properties of the virus during disease progression [4] and contributes to the rapid evolution of HIV-1 toward the establishment of a viral reservoir early after infection [5,6], which impedes HIV-1 cure and vaccine efforts [7,8]. For 80% to 90% of heterosexual transmission cases, it is estimated that a single HIV virus strain is capable of establishing successful transmission and is referred to as a transmitted/founder (TF) viruses [9,10,11,12]. However, if the recipient presents genital ulceration or if transmission occurs via other routes, more than one strain can contribute to the initiation of infection [10,13,14].

The envelope glycoprotein of HIV-1 is a complex trimeric glycoprotein found on the surface of the virus and composed of gp120 subunit spikes linked in a noncovalent interaction with gp41. It is encoded by a viral genome and interacts with host cells and plays an essential role in the virus replication cycle by mediating the fusion between viral and cellular membranes during the entry process [15]. The HIV-1 envelope glycoprotein harbors the transmembrane domain [15]. The HIV-1 envelope structure starts with an N-terminal signal peptide (SP) that guides the nascent Env to the endoplasmic reticulum (ER), where Env synthesis and posttranslational modifications take place [16]. The gp120 glycoprotein is subdivided into five conserved subdomains (C1–C5) and five hypervariable glycosylated loops (V1–V5) [15,17,18,19] involved in HIV-1 pathogenesis and viral escape. HIV-1 gp41 consists of 345 amino acids [20], thus facilitating viral protein and host cell membrane fusion [21,22]. Gp41 has three subdomains [23] and seven functional domains, including the ectodomain that mediates membrane fusion [21,24] and the conserved transmembrane domain (TMD) [25], which acts as a membrane anchor that prevents the release of gp160 into the lumen of the endoplasmic reticulum (ER) [26,27], mediates fusion of the viral envelope with the host cell membrane [15] and enhances cell to cell and virus to cell fusion [28], and the cytoplasmic domain, also called the cytoplasmic tail (CD) [15,29,30]. The latter includes three lentivirus lytic peptides segment 1, 2 and (LLPs) [30,31]. LLPs contribute to the surface expression of Env [32,33], Env incorporation into viral particles [34,35], fusogenicity [36,37] and its localization in lipid rafts [31].

The envelope glycoprotein of HIV-1 is encoded by a viral genome and this protein mediates the first contact with host cells. Even discrete or predominant, an amino acid change (mutations, insertions and deletions) in this viral specific region merit further analysis. Such changes may constitute an important genetic signature developed at the acute stage of infection and selected by TF viruses. They may constitute as keys factors that influence viral fitness and enhancing HIV transmission to new recipient.

The current study characterized the HIV-1 envelope variable region with regard to loop length, N-linked glycosylation sites and the V3-positive net charge. We also screened the Env full-length amino acid sequences to identify mutation patterns occurring in acute/early stages of HIV infection.

## 2. Materials and Methods

### 2.1. Description of Specimens

We used specimens from HIV-1-infected individuals obtained from the Laboratoire de santé publique du Québec (LSPQ) of the Institut national de santé publique (INSPQ) serobank collection (original samples) from first time individual HIV diagnosis before enrollment in any antiretroviral therapy. The majority HIV-1 infected population is men who have sex with men (MSM) and someone injecting drug users (IDU). The LSPQ is the Public Health Laboratory of the province of Quebec, where confirmation testing of HIV infection is performed on all HIV-positive tests in Quebec. All the LSPQ specimens used in this study were residual sera first collected for routine diagnostic purposes between year 1991 and 2015. Specimens deriving from acute HIV-1 infection contained TF viruses (Fiebig stage 1 and 2) and were selected based on p24 antigen-positivity in the absence of HIV-1 antibody [38]. The recent (RC; Fiebig 3 to 5) and chronic (CH) HIV-1-infected individuals’ serum samples were determined by recent testing algorithm (RITA) performed at the LSPQ [39]. The mean duration of recent infection (MRDI) is defined as less than 136 days of infection [39]. Confirmed untreated chronic clade B HIV-1 envelope sequences were also obtained from the Los Alamos sequence databases (LANL) as part of the data set for chronic infection in addition to chronic viruses’ envelope sequences obtained among public health laboratory of Quebec (LSPQ) serobank samples collection. The genetic diversity in the HIV-1 envelope sequence of TF viruses [40] may govern the functional properties and provide clues on viral transmission success and immune evasion [9]. Recent HIV viruses usually correspond to those identified at early stage of infection within the 6 month period [40,41]. For this study, we considered early HIV infection a period within mean duration of recent infection (MDRI) of 136 days [39]. HIV-1 viruses identified at this period were defined as recent viruses. The distribution of the specimens and sequences per category of infection are described in Figure 1 and referred nomenclatures in Appendix A.

### 2.2. HIV-1 RNA Extraction

One hundred (100) microliters of unique serum from each HIV-1-infected subject was used for HIV-1 RNA extraction using BioRobot MDx automated viral RNA extraction. The QIAamp^®^Virus BioRobot^®^ MDx Kit (Qiagen, Valencia, CA, USA) was used. To respect the minimum 350-µL sample volume required for BioRobot automate extraction, we diluted each 100-µL serum sample with 250 µL Dulbecco’s Modified Eagle’s Medium (DMEM; Sigma-Life Science, Oakville, Ontario, CA, USA). Extraction was conducted automatically according to the manufacturer’s protocol. One positive and one negative control sera were always included in each panel of extraction for quality control. Suspension of extracted RNA (approximately 60–80 µL) was immediately used for reverse transcription or stored at −80 °C for reference use.

### 2.3. Reverse Transcription (RT-PCR)

Reverse transcription of the full-length HIV-1 RNA envelope was performed using the following primers: SG3-lo forward and SG3-up reverse [42,43], which cover 4.7 kb fragments of Env length. The extracted RNA (5 µL) was reverse-transcribed in a total volume of 50 µL using the Superscript III One-Step RT-PCR system with Platinium^®^ Taq DNA polymerase ((Invitrogen, Carlsbad, CA, USA), which included 2.0 µL SuperScript™ III RT/Platinum^®^ Taq mix (10 U/µL), 25 µL of 2X reaction mix at a concentration of (05 µM), 2.0 µL of each primer (SG3-lo and SG3-up) at a concentration of (10 µM), 0.5 µL of RNase OUT at concentration of (2 U/µL) completed with13.5 µL of diethylpyrocarbonate (DEPC) water. 

The RNA product (10 µL) was first denatured at 65 °C for 5 min in a thermocycler (Applied Biosystems (ABI) GeneAmp PCR System 9700). The denatured RNA (5 µL) was added to the 45 µL reaction mix containing primers, RNase OUT and the Platinium^®^ Taq DNA polymerase. The reaction mix was then placed in an Applied Biosystems (ABI) thermocycler for cDNA synthesis. The thermal profile was as follow: 53 °C for 30 min and 94 °C for 2 min, followed by 40 cycles at 94 °C for 2 min for denaturing, 55 °C for 30 s for annealing, 68 °C for 4 min for extension and 68 °C for 5 min with a final hold at 4 °C. The PCR product was immediately amplified (nested PCR) or stored at 4–8 °C for future use.

### 2.4. Second Amplification

After RT-PCR, a nested polymerase chain reaction (nested PCR) of the full-length HIV-1 envelope gene (GP160) was amplified using appropriate primers that covered a fragment of 3.10 kb. The following primers were used: Env-up forward (5′-GTTTCTTTTAGGCATCTCCTATGGCAGGAAGAAG-3′), HXB2 nucleotides (nts) position (5957–5983) and Env-lo reverse (5′-GTTTCTTCCAGTCCCCCCTTTTCTTTTAAA AAG-3′), HXB2 nts position (9063–9088). For PCR amplification, cDNA (2 µL) was amplified in a total volume of 50 µL using the Expand™ High Fidelity PCR System (3.5 U/µL) enzyme (Roche Life Sciences, Mannheim, Germany). For the preparation of the reaction mix, (2 µL) of cDNA was added to 0.4 µL of the expand high fidelity enzyme mix, 5.0 µL of the expand high fidelity buffer at (10^×^) concentration, 1.5 µL of dNTPs at concentration of (10 mM) and 2.0 µL for each primer (Env-up and Env-lo) at a concentration of 10 µM) completed with 37.10 µL of diethylpyrocarbonate (DEPC) water.

For amplification in the ABI 9700 thermocycler, the following temperatures were used: 94 °C for 2 min, followed by 45 cycles of denaturing at 94 °C for 15 s, annealing or hybridization at 55 °C for 30 s, and extension or elongation at 68 °C for 2 min, followed by 68 °C for 7 min and a hold at 4 °C. The PCR products were immediately visualized on a 1% agarose gel by electrophoresis and purified using DNA purification kits from QIAGEN and stored at −20 °C before sequencing.

### 2.5. DNA Sequencing and Sequence Assembly

Two µL of input purified DNA was quantified by a Nanodrop and the appropriate concentration was established. In addition, 5 µL (0.2 ng/µL) of input purified DNA was also quantified by iQ™5 Optical System Software, (Bio-Rad Laboratories Ltd. Ontario, Canada) using PicoGreen dsDNA Quantification Reagent). Full-length gp160 of the HIV-1 viral envelope gene was sequenced using MiSeq (Illumina, San Diego, CA, USA) a next generation sequencing (NGS) method with a MiSeq^®^ Reagent Kit (San Diego, CA, USA). The Nextera XT DNA library prep kit (Illumina, San Diego, CA) was used for library preparation and the manufacturer’s protocols were respected. The Illumina MiSeq system was edited using MiSeq Reporter, a bioinformatics data analysis software built into the MiSeq. The workflow using the Nextera XT DNA library kit contains the following steps: (1) tagmentation of genomic DNA, (2) PCR amplification, (3) PCR Clean-up, (4) library normalization and (5) library pooling for MiSeq sequencing were strictly respected. After cycle sequencing, gigabase data provided by Illumina MiSeq were transferred and stored into a securely cloud-based genomics computing environment named BaseSpace Sequence Hub.

### 2.6. Data Management and Analysis

Sequence analysis was performed by cycle-sequencing using Illumina MiSeq. The data produced were viewed by a sequencing analysis viewer (SAV) as recommended by the manufacturer. Individual sequence fragments were assembled using an IVA (iterative virus assembler) and consensus sequences were identified by each specimen. Only the sequences that represented most of 1% of the viral population were retained for subsequent analyses to reduce potential recombination artifacts that may influence viral sequence diversity. Consensus sequences obtained from NGS and IVA assembly obtain from original data and those obtain from Los Alamos HIV-1 sequence database were aligned using Clustal Wallis and conducted in Molecular Evolutionary Genetics Analysis Version 7.0 for Bigger Datasets (MEGA7.0) software (www.megasoftware.net) [44]. A reference “HXB2” (GenBank accession number: K03455.1) envelope sequence, gp160 (amino acids residues 512–856 of full genome numbering) was included in the alignment. Each sequence was also aligned in the blastx homepage with online software (http://blast.ncbi.nlm.nih.gov) and screened to identify potential protein products encoded by a nucleotide query of each sequence. This blastx search ensures that all sequences correspond to the amplified full-length gp160 HIV-1 envelope. All ambiguous or gaps sequences were excluded from subsequent analyses.

We used the online Los Alamos sequence database tools to determine the characteristics of the HIV-1 variable regions (GP120- V1 to V5 loops). This online tool provides results of the HIV N-linked glycosylation site, the loop length and the V3 loop net charge (NC) where we used default setting that have computed with KRH = (+) and DE = (−). The following link was used: https://www.hiv.lanl.gov/content/sequence/VAR_REG_CHAR/index.html.

Determination of the HIV-1 envelope amino acid sequences logo and frequencies for three category of infection stages (TF, RC and CH) were performed using WebLogo Version 2.8 and 3 (http://weblogo.berkeley.edu/logo.cgi) [45,46]. The WebLogo tools generate sequence logo, graphical representations of the patterns within a multiple sequence alignment. HIV-1 envelope sequences subjected to WebLogo analyses (*N* = 231)) were First submitted to multiple sequence alignments using MEGA 7.0 software. The HXB2 Env GP160 sequences were introduced in alignment for numbering purposes [15,29,31,47,48,49]. Aligned envelope sequences for each category of viruses (chronic, transmitted/founder and recent) were then downloaded separately in fasta files. Sequences files for each category of infection were subsequently uploaded individually and analyzed in WebLogo online software. Weblogo analysis identify and counts number of individuals amino acids selected at each position of the Env sequence length (1-856) and report frequency from population. Sequence data of each category of infection were further downloaded in plain text output format that reported the total count of selected amino acids at each position of Env aligned sequence. We finally generated three files of amino acid counts for each of the three category of viruses (TF, RC and CH) and proceeded to statistical comparison; position by position and amino by amino acid, between them and reported the significant difference. We referred to HXB2 Env sequences introduced as reference in MEGA 7.0 multiple sequences alignments to identify the exact position of the amino acids in Env by checking in black the boxes: *without (w/o) gaps*. Localization (sub-regions or domains of Env) for any amino acid changes referred to HXB2 numbering as summarized in Appendix A.

Envelope nucleotide and amino acid sequences for all full-length HIV-1 transmitted founder, recent and chronic viruses were deposited and are available in the GenBank sequence database under the accession MK076153-MK076292. HIV-1 envelope sequence data qualifiers are also available in Appendix A. The background information of selected LANL chronic clade B HIV-1 envelope sequences were reported in the Appendix A.

### 2.7. Statistical Analyses

We used descriptive statistics, the mean and nucleotide composition across HIV-1 envelope gp160 length to estimate the amino acid differences between transmitted founder with recent and chronic sequences using the HXB2 envelope sequence as a reference. Descriptive statistics were performed using proportion and means or median for qualitative and quantitative variables, respectively, as well as ridge plots. The Kruskal Wallis, Wilcoxon and Chi square tests were used to compare the different parameters per type of infection. The Wald test statistics with logistic regression model were also used. Stata version 14, R version 3.5.1 and SPSS version 24 were used as statistical software. A *p*-value less than 0.05 was considered statistically significant. Moreover, the *p*-values were adjusted using the Benjamin Hochberg procedure for multiple comparisons.

### 2.8. Ethics Approval and Consent to Participate

Ethical approval was given by the “Comité d’éthique et de la recherche (CÉR) des Centres hospitaliers affiliés à l’Université de Montréal (CHUM); Number: 2015-5569, CE14-344CA. It was yearly renewed since 2015 by our Institutional review board. All samples were anonymized before application in this study. No nominal information was used for analysis or data management. This manuscript did not contain any individual data in any form whatsoever to publish.

## 3. Results

A total (*N*) of 757 specimens from acute and early HIV-1 infections based respectively on EIA-p24 antigen positive, Western blot (WB) negative and WB positive with the presence of HIV-1 antibody and qualified by a recent infection testing algorithm (RITA) [37,39] (Figure 1). Chronic clade B HIV-1 viruses envelopes sequences were selected from Los Alamos HIV sequence databases and constitute a part of chronic sequences that were obtained from the LSPQ serobank collection (Figure 1, Appendix A).

From acute HIV-1-infection serum samples (*N* = 469) classified as TF viruses, we obtained 98 consensus individual clade B HIV-1 envelope sequences after a molecular evolutionary genetic analysis. Five of the one hundred tree sequences were excluded because they had short envelope amino acids sequences lengths (<856 bp) after MEGA 7 multiples sequences alignment. The nested-RT-PCR success rate of acute infection samples was 23% (102/469), as presented in Figure 1.

From the total of early HIV-1 infection samples (*N* = 240) where viruses identified as classified as recent HIV viruses, twenty-eight (28) HIV-1 consensus envelope sequences were obtained (Figure 1). This result corresponded to an RT-PCR amplification success rate of 15% (36/240). Eight (8) non-B HIV-1 envelope sequences were excluded for molecular analysis (Figure 1).

Of forty-eight (48) chronic HIV-1 infection samples collected from LSPQ serobank samples collections, only two HIV-1 envelopes sequences were finally obtained after analysis, which demonstrated a fair result after HIV RNA amplification (4%; Figure 1).

The repartition of clade B HIV-1 envelope sequences (one sequence per individual) in total is as follow: chronic (CH): 45.46% (*N* = 105 include); TF viruses: 42.42% (*N* = 98 include, *N* = 4 non-B HIV-1 were exclude) and recent (RC): 12.12% (*N* = 28 were include, N=8 non-B HIV-1 were excluded) as shown in Figure 1 and Appendix A. A total of 105 HIV-1 B chronic envelope sequences included two HIV-1 envelope sequences of LSPQ serobank samples collections and LANL chronic HIV-1 clade B envelope sequences (Figure 1). The background information of LANL selected chronic HIV-1 envelope sequences is available in Appendix A.

Thus, a total of 231 clade B HIV-1 full-length consensus envelope sequences were included in this analysis.

### 3.1. Characteristics of HIV-1 Envelope Variable Regions

The first objective of the current study was to determine the characteristics of the HIV-1 variable regions, including the number of N-glycosylation sites, the loop length and the V3 loop positive net charge between TF, RC and CH viruses. As presented in Figure 2d and Appendix A, the differences in the numbers of N-glycosylation sites of the HIV-1 Env GP120 loop 3 (V3) were statistically significant using the Kruskal–Wallis test and Wald test with logistic regression model. This concerns CH (median/range: 2 (2,2)) and TF (median/range: 2 (2,2)),(OR = 0.58, 95% CI (0.36–0.93), *p* = 0.026), between RC (median/range: 2 (2,3) and TF: 2 (2,2) (OR = 0.37, 95% CI (0.19–0.73), *p* = 0.004 and CH (median/range: 2 (2,2) and RC (median/range: 2 (2,3), (OR = 2.03, 95% CI (0.98–4.21), *p* = 0.05).

The difference in the HIV-1 Env GP120 loop 5 (V5) lengths was statistically significant using the Kruskal–Wallis test and Wald test with logistic regression model (Figure 3f and Appendix A). These differences concerned RC (median/range: 15 (13, 15) and TF (median/range: 13 (12, 14)), OR = 0.65, 95% CI (0.49–0.86), *p* = 0.003 and between CH (median/range: 15 (12, 14)) and RC (median/range: 15 (13, 15), OR = 1.44, 95% CI (1.12–1.86), *p* = 0.004.

The positive net charge of the HIV-1 Env GP120 loop 3 (V3) was also statistically significant between CH (median/range: 5 (3, 6), RC (median range: 4 (3, 5.5) and TF (median/range: 4 (3, 5), *p* = 0.040 and specifically, between CH (median/range: 5 (3, 6), and TF (median/range: 4 (3, 5), viruses (OR = 0.82, 95% CI (0.69–0.98), *p* = 0.038) using the Kruskal–Wallis test and Wald test with logistic regression model (Figure 4; Appendix A).

The box represents a density plot of the V3 positive net charge. The top (green), middle (blue) and bottom (yellow) represent the TF, RC and CH viruses’ V3 sequences net charge, respectively. The X-axis represents the number of charges for the HIV-1 Env gp120V3 loop and the Y-axis represents the density of sequence charges of HIV-1-infected individuals for the CH, RC and TF viruses respectively. As shown in Figure 4, the difference of the V3 positive net charge was significant between CH, RC and TF, *p* = 0.04. Importantly, the difference in HIV-1 Env V3 loop net charge was statistically significant between CH and TF viruses, *p* = 0.03 using Wald with regression logistic model. No significant difference was observed between RC and TF, *p* > 0.05.

### 3.2. Clade B HIV-1 Envelope Amino Acids Signatures Associated to Transmitted/Founders and Recent Viruses Compared to Chronic

The second objective of this study was to screen full-length HIV-1 envelope sequences to identify genetic characteristics (mutation patterns) associated with transmitted/founder and recent viruses compared to chronic ones. As presented in Figures 6, 7 and Table 1, two genetic signatures were identified.

The first significant amino acids enrichment difference between CH and TF was observed in the HIV-1 envelope gp41 cytoplasmic tail, specifically in the Lentivirus Lytic peptide 1 (LLP-1). It concerns a substitution of an arginine (R) by an isoleucine (I) at position 841 (R841I) in reference to HXB2 Env sequence numbering (Figure 5, Table 1).

The isoleucine was highly enriched in TF viruses LLP-1 domains, 33.68% (32/95), compared to CH virus LLP-1 domains, 9.52% (10/105), OR = 0.2, 95% CI (0.09–0.44), *p* < 0.00001 using the hi-squared (Chi²) test.

The second genetic signature was identified in the HIV-1 envelope signal peptide (SP). It concerns a substitution of lysine (K) by an isoleucine at position six of HXB2 numbering (Figure 6, Table 1). The substitution of lysine (K) for isoleucine (I; K6I) in the Env SP at position six was highly enriched in chronic viruses, 79.04% (83/105) compared to TF viruses, 53.60% (52/97), OR = 3.26, 95% CI (1.76–6.02), *p* = 0.0001 using the chi-squared (Chi^2^) test.

Others significant amino acids mutation patterns that distinguish TF from CH HIV-1 envelope sequences were also found less significant amino signatures in the GP120 C1 VI, V5 loops and GP 41 fusion peptide (FP), Kennedy Epitope (KE), loop and Fusion peptide proximal region (FPPR; Table 1).

### 3.3. HIV-1 Envelope Genetic Signatures Among Transmitted/Founder and Recent Viruses Compared to Chronic

Four important genetics signatures were also identified when combining TF and RC compared to chronic viruses (Figure 7, Figure 8, Figure 9 and Figure 10). The first one was localized in the GP120 V1 loop at position 153 (Figure 7) and did not constitute a change. However, it identified a high enrichment of glutamic acid (E; 153E) in CH viruses, 89.42% (93/104) and 65.60% (82/105) in RC (+TF) ones, OR = 4.43, 95% CI (2.16–9.05), *p* = 0.000001 using the Chi^2^ test.

The second was found in the Env signal peptide at position 24 and identifies a substitution of a methionine (M) by an Isoleucine (I; M24I; Figure 8). The isoleucine mutation was enriched at 36.53% (38/104) for CH and 12.08% (16/125) for RC (+TF), OR = 3.92%, 95% CI (2.04–7.53), *p* = 0.00001 using the Chi^2^ test.

The third signature was localized at position 621 of HIV-1 Env GP41 loop domain. It consisted of glutamine (Q) substitution by aspartic acid (D; Figure 9). The aspartic acid was enriched at 15.38% (16/104) for CH and 41.60%, (52/125) for RC (+TF), OR = 0.25%, 95% CI (0.13–0.48), *p* = 0.00001 using the Chi^2^ test.

The last amino acid mutation patterns that distinguish chronic from recent viruses were localized in the HIV-1 Env GP 41 cytoplasmic tail specifically at position 751 (Figure 10). Specifically, they were localized between the NF-κB activation (NA) and the highly immunogenic region, also called Kennedy Epitope (KE; Figure 10). It consisted of an aspartic acid (D) substitution by a valine (V; D751V). The valine was enriched at 60.95% (64/105) for CH and 31.20% (39/125) for RC (+TF), OR = 0.25%, 95% CI (1.99–5.92), *p* = 0.00001 using the Chi^2^ test.

The complete profile of statistically significant clade B HIV-1 envelope amino acid genetic signatures that distinguish recent from chronic viruses is summarized in Table 2.

## 4. Discussion

The main objective of the current study was to determine the characteristics of the clade B HIV-1 envelope variable loop in term of sequences length, number of N-glycosylation sites and net charge. It also aimed at identifying the principal amino acid signatures associated with TF and RC founder virus strains compared to chronic viruses. The TF and RC HIV-1 viruses envelopes glycoproteins mutations patterns determine the success of viral transmission and its evolution during HIV-1 infections. Identifying such genetic signatures may help improve HIV-1 prevention and inform vaccine design. The current study included 103 untreated HIV-1 clade B HIV-1 consensus envelope sequences from different cohorts available in the Los Alamos sequence databases (Figure 1), in addition to two sequences derived from LSPQ serobank chronically infections. To limit the selection bias in LANL sequences, we carefully identified consensus sequences (one/patient) from clearly untreated chronically HIV-1-infected individuals from the North America region (United States of America and Canada) that had been previously included in published articles [1,9]. We failed to obtain more HIV-1 chronic envelope sequences from all study participants derived from LSPQ serobank collections in order to make comparisons between TF and CH derived in the same context. This was due to the lower amplification success rate obtained in this study for those samples. 

Multiple factors may have affected HIV-1 envelope amplification success rate including sample quality such as the long-term storage, viral RNA extraction procedures, primers and enzymes as well as the viral loads of infected individuals (VL < 20,000 copies/ml). Depending on the length the HIV-1 genome to be amplified and specifically for Env gene, the procedure is known to be challenging [9,50,51,52].

### 4.1. Clade B HIV-1 Envelope Variable Loop Characteristics

The first objective of the current study was to characterize HIV-1 TF viruses envelope variable regions, which include the V1/V2, V3, V4 and V5 loop lengths, their number of N-linked glycosylation sites and the V3 loop positive net charge.

Our results show that the V3 loop numbers of N-glycosylation sites of TF viruses were significantly less glycosylated than the chronic ones (Figure 2 and Appendix A). The Env V3 loop of TF viruses were less positively charged than chronic viruses. (Figure 4 and Appendix A). This observation confirms earlier findings of a decreased positive net charge of TF viruses V3 loop sequences compared to chronic [53,54,55,56,57,58]. The positive net charge of HIV-1 envelope hypervariable loop three modulated the viral phenotype and tropism [59] at different stages of infection. The lower decreased charge of TF viruses V3 loop may constitute a regulating factor of viral phenotype during transmission.

In this study, The V1/V2 loop length and number of N-glycosylation sites did not differ between TF and CH HIV-1 viruses envelopes identified by earlier studies [60]. A shorter V1/V2 length and a fewer number of N-glycosylation sites have been associated to TF viruses in previous studies [60]. Most of these characteristics have been observed for clades A, C and D of HIV-1 [1,14,58,61]. This could reflect a difference among clades, as our study compared clade B HIV-1 Env V1 and V2 loops [14,57,58].

We also observed that HIV-1 Env GP120 loop 5 (V5) length of TF viruses was significantly shorter than RC and CH viruses (Figure 3 and Appendix A). The V5 loop has been found to be necessary for viral structure integrity maintenance, negatively affected virus assembly and virus entry [62] and constituted neutralizing determinants recognized by broadly neutralizing monoclonal antibodies [63,64]. It also participates in CD4 binding sites (CD4bs) formation [65]. The shorter loop length of TF compared to RC and CH viruses suggests that V5 sequence loops length modeling at the acute stage of infection plays an important role in the virus transmission process and subsequently to disease progression.

### 4.2. Clade B HIV-1 Envelope Amino Acids Signatures Associate to Transmitted/Founder and Recent Viruses

The second objective of this study was to identify specific mutation patterns across the HIV-1 envelope that may be considered as a genetic signature of TF viruses. We first compared the TF viruses envelope sequences derived of acutely HIV-1-infected individuals (Fiebig stage 1 to 2) [66,67] with those from chronically infected individuals. The first important point mutation identified consisted of substitution of an arginine (R) by an isoleucine (I) at position 959 of alignment and referred to HXB2 numbering to position 841 (R841I; Figure 5). This mutation was localized in the C-terminal of the cytoplasmic (CT/D), specifically in the lentivirus lytic peptide segment 1 (LLP-1) [68,69]. The cytoplasmic tail or domain of GP41 is important for HIV-1 replication and pathogenesis by regulating rapid clathrin-mediated endocytosis that induces low levels of Env expression on cell surface [30,70,71]. This phenomenon contributes to limiting humoral immune pressure to HIV-1 [30]. It is also known that GP41 CT contributes to Env incorporation into virions by interacting with viral matrix protein [37] and also for cellular-transcription factor NF-κB activation [30]. The CTs of HIV-1 of GP41 have also been shown to have an impact on gp120 and ECD conformation and mutations in this domain also impacted recognition and neutralization of antibody [15,32,72].

The R841I signature associated to clade B HIV-1 TF viruses was localized at position 841 of the GP41 cytoplasmic tail in this LLP-1. It was reported that the LLP-1 mutations affect Env association with lipid rafts [31,70] and reduce Env incorporation, infectivity and the replication process for certain viral phenotypes [73]. The results of our current study reveal the selection of isoleucine by TF viruses (Table 1), which may contribute to HIV-1 gp41 CT functions. Earlier studies have highlighted the importance of gp41 cytoplasmic tail domain (CD) in HIV-1 in transmission and pathogenesis [30]. Lee, S. F. et al. (2002) previously demonstrated that a single deletion of one of the two adjacent valine residues located at position 832 and 833 and Ile-830, Ala-836 and Ile-840 significantly contributed to the reduction of Env steady-state expression [74]. The R841I substitution identified in the acute stage of infection of our study may constitute a key factor to enhance LLP-1 functions. It would be necessary to evaluate its functional implications in the HIV-1 transmission process and viral replication.

The second important amino acid signature identified was K6I (within the signal peptide), highly enriched in chronic vs. TF viruses (Figure 6, Table 1). The negative selection of this mutation for the TF viruses may represent a strategy for virus resistance to early immune responses. Gnanakaran S, et al. 2011 [1] showed that a histidine signature at position 12 (H12) in the signal peptide was highly enriched in TF viruses compared to chronic HIV-1 envelope sequences. The histidine amino acids that were normally located at position 12 were substituted by arginine (R) or proline (P) during acute infection [1,75]. This H12 signature was found to increase envelope incorporation in pseudoviruses in vitro [75,76]. The current study identified and highly selection of the isoleucine (K6I) associated to chronic compared to TF viruses (Figure 6, Table 1). This mutation constituted an amino acid signature enriched during disease progression over chronic infection.

The HIV-1 envelope signal peptide plays an important role in virus interaction with host cells during transmission and its evolution toward the chronic stage. It contributes to increased Env gp120 transport and the secretion and expression of Env on the cell membrane surface [77]. As reported by previous studies, a natural variation in the N- terminal signal peptide (SP) of the HIV envelope significantly impacts the antigenicity and molecular mass of mature gp120 and its glycosylation and interaction with DC-SIGN [78]. The SP is also likely subjected to antibody-mediated immune pressure [77]. Compared to the Gnanakaran et al. (2011) study, phylogenetic analysis methods and the numbers of sequence datasets used for amino acid signature estimate may explain the different results. The Gnanakaran et al. study used consensus and corrected phylogenetic tree analyses [1,76] for amino acid signature estimates, whereas our study used the WebLogo online-based application to map and determine the amino acid estimates. In addition, the definition of the TF viruses was also different. The current study considered TF viruses as the sequences of HIV-1-infected individuals sampled during acute infection Fiebig 1 to 2 stages [40,41], whereas Gnanakaran et al. (2011) considered as recent viruses those identified at early stage of HIV-1 infection covering the Fiebig stages 2 to 5 [1,76].

We believed that identifying the HIV-1 envelope genetic signature very early after infection might lead to a better identification of important genetic polymorphism observed during disease transmission.

Therefore, we compared chronic virus envelope sequences with those of TF and recent ones to determine a specific mutation pattern. Four significant HIV-1 envelope amino acid signatures were identified (Figure 7, Figure 8, Figure 9 and Figure 10 and Table 2). Three were highly associated to chronic viruses and one to recent ones. The mutations patterns associated with chronic viruses consisted of glutamic acid (E) at position 153 in the V1 loop (153E), a methionine (M) substitution by isoleucine at position 24 in the signal peptide (M24I) and aspartic acid (D) substitution by a valine (V) in the cytoplasmic tail (D751V; Figure 7, Figure 9 and Figure 10 and Table 2). The amino acid signature associated with recent and TF viruses together was localized in the GP41 ectodomain, specifically in the loop domain (Figure 9 and Table 2). It consisted of glutamine (Q) substitution by aspartic acid (D) at position 621 (Q621D). The HIV-1 envelope genetic signatures identified for chronic and recent viruses constituted results of accumulated mutations during disease progression.

In summary, we identified an important HIV-1 envelope amino acid genetic signature associate to the GP41 cytoplasmic tail, specifically in the lentivirus lytic peptides associated TF compared to chronic viruses. It would be interesting to conduct phenotypic studies to further evaluate the role of isoleucine substitution in viral Env function (R841I) including others frequents mutations. Overall, this study provided new evidence related to genetic characteristics of HIV-1 envelope sequences associate with clade B TF, RC and CH viruses. As other genetic analysis, the Weblogo would have been impacted results of sequence genetic profiles if different groupings were considered. Careful verification to ensure that all sequences have the same lengths and do not contain gaps could be considered.

## 5. Conclusions

The current study identified the presence of different point mutations patterns in the HIV-1 envelope, specifically in the GP41 cytoplasmic tail lentivirus lytic peptide segment 1 (gp41 CT_LLP-1) significantly associated with TF viruses. The LLP-1 domains of GP41 CT play an important role in the virus replication and pathogenesis. The R841I mutation identified in this segment may be considered as specific genetic signature, as well as its phenotypic properties during HIV-1 transmission merits further study. The HIV-1 transmission is complex, multifactorial and this mutations profiles identified could not be the only contributing factors to the disease transmission. But, understanding and identifying such early envelope molecular determinants may provide clues for the design of an HIV vaccine.

## Figures and Tables

**Figure 1 viruses-11-01012-f001:**
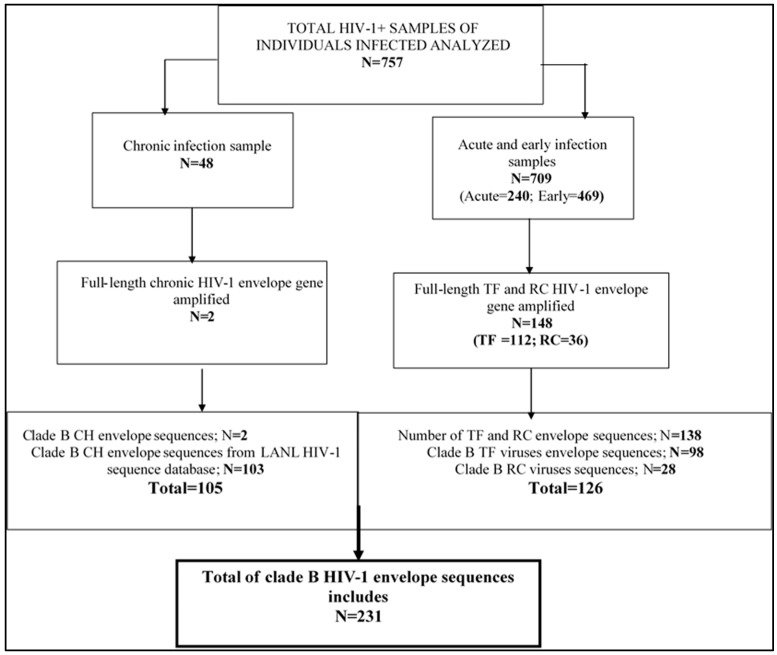
Flow chart showing the numbers of HIV-1 positives samples analyzed and consensus envelope sequences of HIV-1-infected individuals include in the study. From a total of 243 sequences obtained, 95% (231) of subtype B HIV-1 sequences were included in the current study. The non-B HIV-1 subtype envelope sequences representing 8.7% (four for transmitted/founder (TF) and eight for recent (RC) viruses) of the LSPQ serobank HIV-1 positive samples were excluded for subsequent genetic analysis because they may have influenced the results. The HIV-1 envelope sequences of transmitted/founder (TF) viruses derived from acute infection, recent viruses (RC) for early infection and chronic viruses for chronic infection.

**Figure 2 viruses-11-01012-f002:**
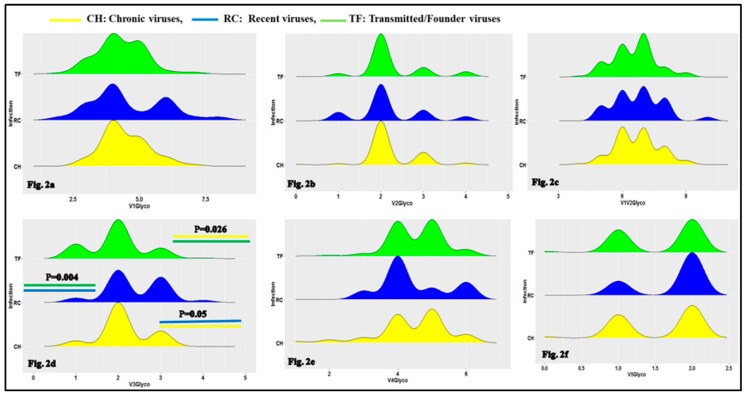
Ridge plot comparing the HIV-1 envelope variable loop number of N-glycosylation sites between TF, RC and chronic (CH) viruses. The boxes represent a density plot of number of N-glycosylation sites for: Env V1 loop (**a**), Env V2 loop (**b**), Env V1V2 loop (**c**), Env V3 loop (**d**), Env V4 loop (**e**) and Env V5 loop (**f**) for CH, RC and TF viruses respectively. In box, the top (green), middle (blue) and bottom (yellow) represent respectively number of N-glycosylation sites for CH, RC and TF viruses envelope sequences respectively. The X-axis represents sequence loops number of N-glycosylation sites and the Y-axis the density of sequences number of N-glycosylation sites for each timeline category of viruses (CH, RC and TF). As shown in Figure 2d, the differences in Env V3 loop numbers of N-glycosylation sites between CH and TF, *p* = 0.026; RC and TF, *p* = 0.004 and CH and RC, *p* = 0.05 were statistically significant using Wald test with logistic regression model.

**Figure 3 viruses-11-01012-f003:**
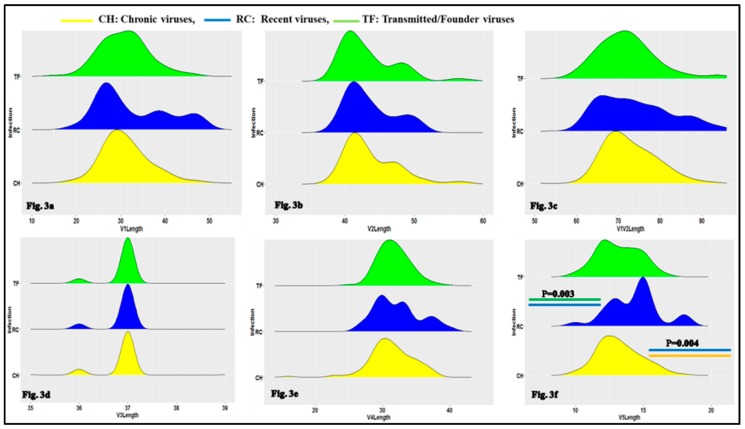
Ridge plot comparing the clade B HIV-1 envelope variable loop length between CH, RC and TF viruses. The boxes represent the density plot of: Env V1 loop lengths (**a**), Env V2 loop lengths (**b**), Env V1V2 loop length (**c**), Env V3 loop length (**d**), Env V4 loop length (**e**) and Env V5 loop length (**f**) for CH, RC and TF viruses respectively. For each box, the top (green), middle (blue) and bottom (yellow) represent, respectively, TF, RC and CH viruses sequences. X-axis presents sequence loop lengths and the Y-axis the loop length density for each timeline category of viruses (CH, RC and TF). As presented in Figure 2f, the differences in the HIV-1 Env GP120 V5 loop lengths between RC and TF, *p* = 0.003 and CH and RC, *p* = 0.004 are statistically significant using Wald Test with regression logistic model.

**Figure 4 viruses-11-01012-f004:**
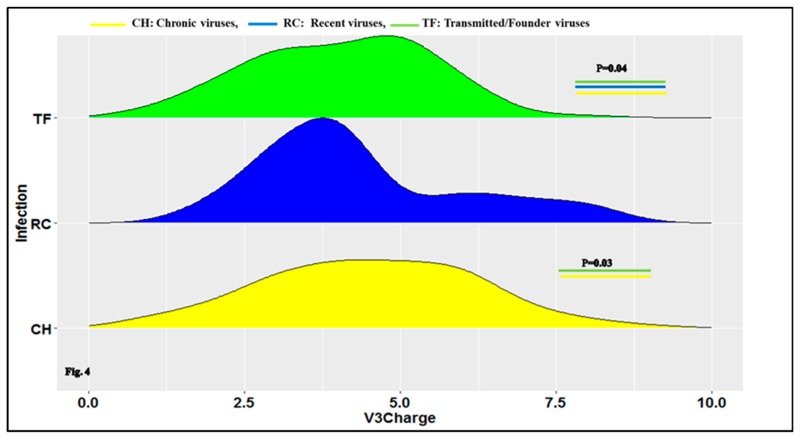
Ridge plot comparing the clade B HIV-1 envelope variable region gp120 loop 3 (V3) net positive between CH, RC and TF viruses.

**Figure 5 viruses-11-01012-f005:**
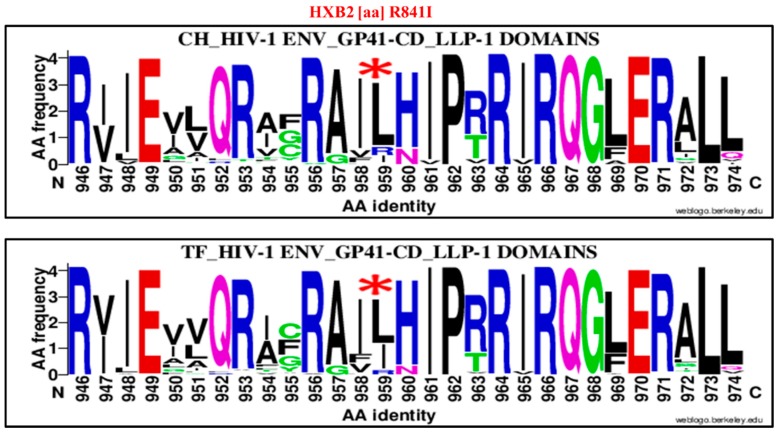
Genetic signature identified under the HIV-1 envelope GP41 Lentivirus Lytic peptide 1 (LLP-1) associate to clade B HIV-1 TF viruses compared to chronic (CH) using WebLogo. The X axis represents amino acids (AA) composing the LLP-1 sequence (direction N to C). The Y axis represents the normalized AA frequency identified at each position of the LLP-1 sequence for each category of infection. The top line box represents the chronic (CH) viruses envelope GP41 LLP-1 sequence (*N* = 105) and the bottom line for TF viruses (*N* = 98). As indicated for Weblogo analysis, the overall height of each stack indicates the sequence conservation at that position (measured in bits), whereas the height of symbols within the stack reflects the relative frequency of the corresponding amino or nucleic acid at that position. The isoleucine (I) amino acid signature was localized at position 959 of alignment (HXB2 position R841I) and identified by a red asterisk.

**Figure 6 viruses-11-01012-f006:**
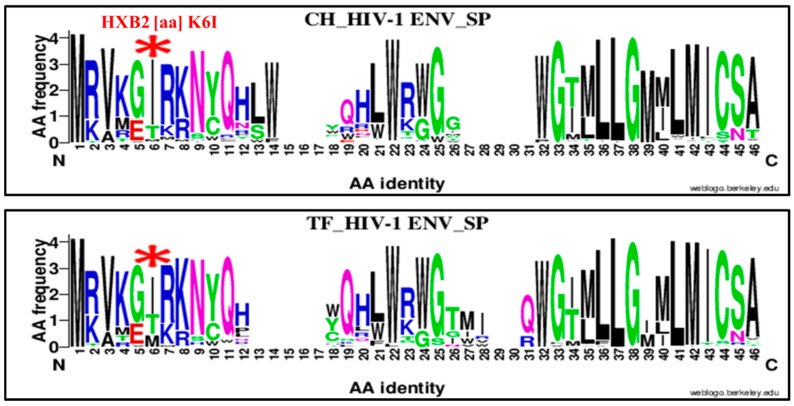
Genetic signature identified under the HIV-1 envelope signal peptide (SP) associate to clade B HIV-1 chronic compared to TF viruses using WebLogo. The X axis represents sequences and amino acids (AA) identities composing the SP (direction N to C). The X axis represents amino acids (AA) composing the LLP-1 sequence (direction N to C). The Y axis represents the normalized AA frequency identified at each position of the SP sequence for each category of infection. The top line box represents chronic (CH) viruses envelope SP sequence (*N* = 105) and the bottom line for TF viruses (*N* = 98). As indicated for Weblogo analysis, the overall height of each stack indicates the sequence conservation at that position (measured in bits), whereas the height of symbols within the stack reflects the relative frequency of the corresponding amino or nucleic acid at that position. The isoleucine (I) amino acid signature was localized at position six of alignment (HXB2 position K6I) and identified by a red asterisk.

**Figure 7 viruses-11-01012-f007:**
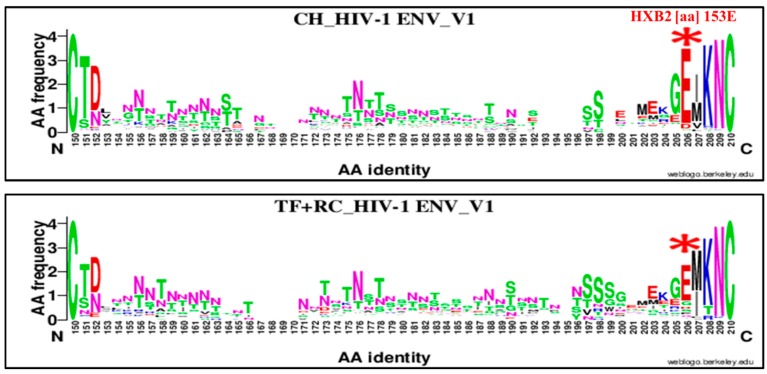
Genetic signature identified under the HIV-1 envelope GP120 V1 loop associate to clade B HIV-1 chronic compared to recent viruses. The X axis represents amino acids (AA) composing the V1 loop sequence (direction N to C). The Y axis represents the normalized AA frequency identified at each position of the V1 sequence for each category of infection. The top line box represents chronic virus envelope V1 loop sequences (*N* = 105) and the bottom for RC viruses within 136 days MDRI and TF (*N* = 126). As indicated for Weblogo analysis, the overall height of each stack indicates the sequence conservation at that position (measured in bits), whereas the height of symbols within the stack reflects the relative frequency of the corresponding amino or nucleic acid at that position. The glutamic acid (E) amino signature was localized at position 206 of alignment (HXB2 position 153E) and was identified by a red asterisk.

**Figure 8 viruses-11-01012-f008:**
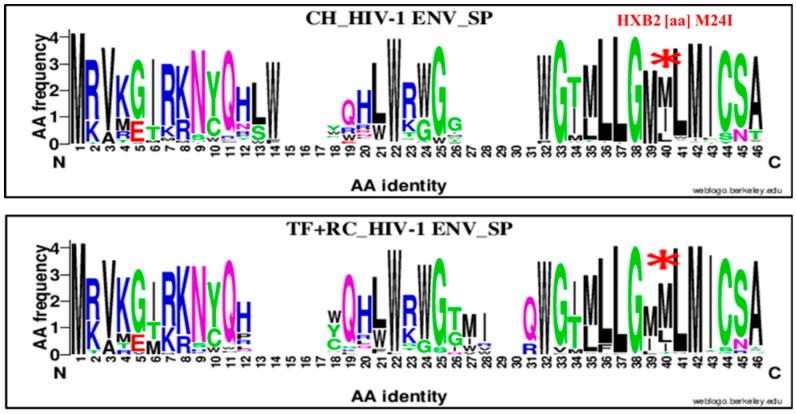
Genetic signature identified under the HIV-1 envelope signal peptide (SP) associate to clade B HIV-1 chronic compared to recent viruses. The X axis represents amino acids (AA) composing the Env SP sequence (direction N to C). The Y axis represents the normalized AA frequency identified at each position of the SP sequence for each category of infection. The top line box represents chronic virus envelope SP sequences (*N* = 105) and the bottom for RC viruses within 136 MDRI and TF (*N* = 126). As indicated for Weblogo analysis, the overall height of each stack indicates the sequence conservation at that position (measured in bits), whereas the height of symbols within the stack reflects the relative frequency of the corresponding amino or nucleic acid at that position. The isoleucine (I) amino signature was localized at position 40 of alignment (HXB2 position M24I) and is identified by a red asterisk.

**Figure 9 viruses-11-01012-f009:**
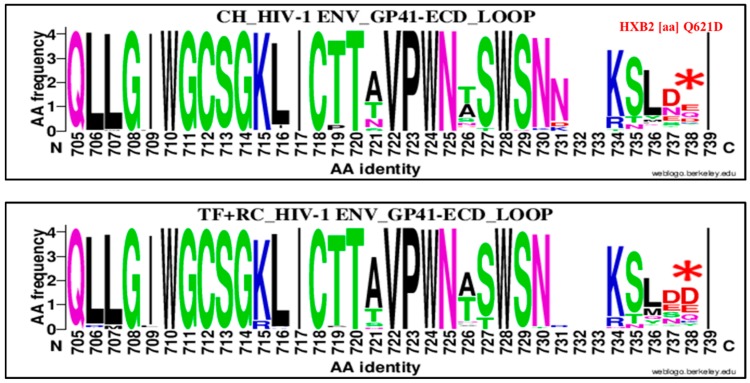
Genetic signature identified under the HIV-1 envelope GP41 cytoplasmic tail (CT or CD) loop domain associate to clade B HIV-1 chronic compared to recent viruses. The X axis represents amino acids (AA) composing the GP41 CT loop sequence (direction N to C). The Y axis represents the normalized AA frequency identified at each position of the GP41 CT loop sequence for each category of infection. The top line box represents chronic virus envelope GP41 CT loop sequences represented individuals (*N* = 105) and the bottom for RC viruses within 136 MDRI and TF (*N* = 126). As indicated for Weblogo analysis, the overall height of each stack indicates the sequence conservation at that position (measured in bits), whereas the height of symbols within the stack reflects the relative frequency of the corresponding amino or nucleic acid at that position. The aspartic acid (D) amino signature was localized at position 738 of alignment (HXB2 position Q621D) and identified by a red asterisk.

**Figure 10 viruses-11-01012-f010:**
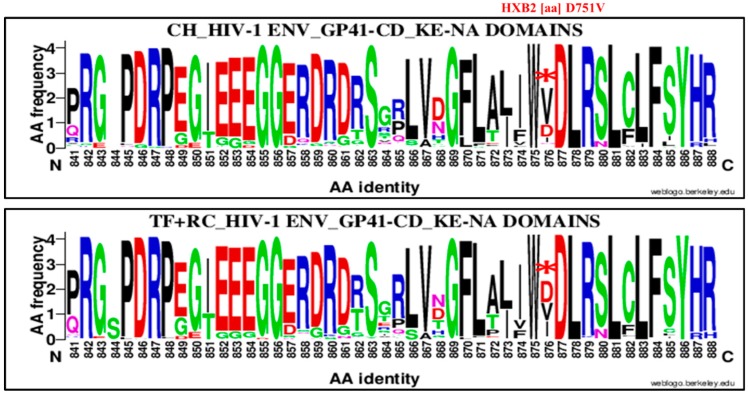
Genetic signature identified under the HIV-1 envelope GP41 cytoplasmic tail (CT or CD) between NF-κB activation (NA) and Kennedy Epitope (KE) domains associate to clade B HIV-1 chronic compared to recent viruses. The X axis represents amino acids (AA) composing the GP41 CT NA and KE sequence (direction N to C). The Y axis represents the normalized AA frequency identified at each position of the GP41 CT, NA and KE sequence for each category of infection. The top line box represents chronic virus envelope GP41 CT NA and KE sequences represented individuals (*N* = 105) and the bottom for RC viruses within 136 MDRI and TF (*N* = 126). As indicated for Weblogo analysis, the overall height of each stack indicates the sequence conservation at that position (measured in bits), whereas the height of symbols within the stack reflects the relative frequency of the corresponding amino or nucleic acid at that position. The valine (V) signature was localized at position 876 of alignment and position 751 referring to HXB2 numbering (D751V) and identified by a red asterisk.

**Table 1 viruses-11-01012-t001:** Summary statistics of the important amino acids change among the HIV-1 envelope sequences of chronic and transmitted/founder viruses.

HXB2		Env		Amino Acids		CH			TF			Chi^2^ Test		
Position	Amino Acids	Subregion/Domain	Alignment Position	Change	Genetic Signature	YES	NO	Total	YES	NO	Total	OR	95% CI	B-H adjusted *p*-values
**841**	**R**	**LLP-1**	**959**	**I**	**R841I**	**10**	**95**	**105**	**32**	**63**	**95**	**0.2**	**0.09–0.44**	**0.00001**
**6**	**K**	**SP**	**6**	**I**	**K6I**	**83**	**22**	**105**	**52**	**45**	**97**	**3.26**	**1.76–6.02**	**0.0001**
62	D	C1	81	E	D62E	20	85	105	4	94	98	5.52	1.89–16.04	0.006
514	G	ECD	628	T	G514T	9	22	31	26	12	38	0.18	0.06–0.52	0.006
24	M	SP	40	I	M24I	38	66	104	16	81	97	2.91	1.50–5.64	0.006
743	R	HIR/KE	865	R	743R	45	60	105	62	35	97	0.42	0.24–0.74	0.008
153	E	V1	206	E	153E	93	11	104	71	26	97	3.09	1.44–6.59	0.008
744	R	HIR/KE	862	R	744R	78	27	105	53	44	97	2.39	1.32–4.32	0.008
717	F	HIR/KE	834	F	717F	86	19	105	62	35	97	2.55	1.34–4.85	0.008
717	F	HIR/KE	834	L	F717L	19	86	105	35	62	97	0.39	0.20–0.74	0.008
154	I	V1	207	M	I154M	35	70	105	52	46	98	0.44	0.25–0.77	0.008
744	R	HIR/KE	862	T	R744T	15	90	105	30	67	97	0.37	0.18–0.74	0.008
841	R	LLP-1	959	L	R841L	81	24	105	56	39	95	2.35	1.27–4.31	0.009
621	Q	GP 41 Loop	738	D	Q621D	16	88	104	31	67	98	0.39	0.20–0.77	0.009
464	L	V5	566	N	L464N	24	45	69	42	31	73	0.39	0.20–0.77	0.009
543	Q	FPPR	657	Q	543Q	63	42	105	76	22	98	0.43	0.23–0.79	0.009

Table 1 presents the important HIV-1 envelope sequence polymorphisms of chronically infected individuals compared to TF viruses of individuals infected that are considered as genetic signatures. Bold typeface define the important Env genetic signatures identified and discussed in manuscript. The gray character referred to sequences alignment position of amino acid change. Abbreviations: V1—variable loop 1, CP—cytoplasmic domain/tail, EC—endocytosis domain, SP—Signal peptide, C2—constant domain 2, C3—constant domain 3, ECD-Loop—Ectodomain-loop region, CP-KE—cytoplasmic domain-Kennedy epitope, ECD-CHR—Ectodomain-C-hepta-repeat, CP-LLP-1—Cytoplasmic Tail-Lentivirus Lytic peptide 1. MSD—Membrane-spanning domain.

**Table 2 viruses-11-01012-t002:** Major amino acid signatures among HIV-1 envelope sequences between chronic and recent viruses.

HXB2		Env		Amino Acid		CH			TF + RC			Chi^2^ test		
Position	Amino acid	Subregion/Domain	Alignment position	Change	Genetic signature	YES	NO	TOTAL	YES	NO	TOTAL	OR	95% CI	B-H Adjusted *p* value
**153**	**E**	**V1**	**206**	**E**	**153E**	**93**	**11**	**104**	**82**	**43**	**125**	**4.43**	**2.16, 9.05**	**0.000001**
**24**	**M**	**SP**	**40**	**I**	**M24I**	**38**	**66**	**104**	**16**	**109**	**125**	**3.92**	**2.04, 7.53**	**0.00001**
**621**	**Q**	**GP 41 CT Loop**	**738**	**D**	**Q621D**	**16**	**88**	**104**	**52**	**73**	**125**	**0.25**	**0.13, 0.48**	**0.00001**
**751**	**D**	**CT (HIR/KE)**	**876**	**V**	**D751V**	**64**	**41**	**105**	**39**	**86**	**125**	**3.44**	**1.99, 5.92**	**0.00001**
6	K	SP	6	I	K6I	83	22	105	69	56	125	3.06	1.70, 5.48	0.0003
33	K	C1	49	Q	K33Q	39	55	94	22	102	124	3.28	1.78, 6.06	0.0003
717	F	EC (YSPL) and HIR/KE	834	F	F717F	86	19	105	73	52	125	3.22	1.75, 5	0.0003
717	F	EC (YSPL) and HIR/KE	834	L	F717L	19	86	105	52	73	125	0.31	0.16, 0.56	0.0003
747	R	HIR/KE NA	865	R	R747R	45	60	105	85	40	125	0.35	0.20, 0.60	0.0003
132	T	V2	151	T	T132T	86	19	105	75	51	126	3.07	1.67, 5.64	0.0004
154	I	V1	207	M	I154M	35	70	105	73	53	126	0.36	0.21, 0.62	0.0004
360	I	C3	441	V	I360V	52	52	104	33	93	126	2.81	1.62, 4.88	0.0004
737	R	HIR/KE	862	T	R737T	15	90	105	44	81	125	0.3	0.16, 0.58	0.0006
841	R	LLP-1	959	I	R841I	10	95	105	35	87	122	0.26	0.12, 0.55	0.0006
744	R	HIR/KE	862	R	R744R	78	27	105	65	60	125	2.66	1.52, 4.65	0.0009
236	T	Loop D	312	S	T236T	9	95	104	33	93	126	0.26	0.12, 0.58	0.001

Table 2 presents the statistically significant clade B HIV-1 envelope sequence polymorphisms among chronic compared to recent viruses of individuals HIV-1 infected that are considered as genetic signatures. Bold typeface define the important Env genetic signatures identified and discussed in manuscript. The gray character referred to sequences alignment position of amino acid changes. Abbreviations: V1—variable loop 1, CP—cytoplasmic domain/tail, EC—endocytosis domain, SP—Signal peptide, C—constant domain 2, C3—constant domain 3, ECD-Loop—Ectodomain-loop region, CP-KE—cytoplasmic domain-Kennedy epitope HIR-KE— Highly Immunogenic region (HIR) also called Kennedy epitope. ECD-CHR—Ectodomain-C-hepta-repeat, CP-LLP-1—Cytoplasmic tail-lentivirus lytic peptide 1.

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
