# Peer review of "HIV-1 Envelope Glycoprotein Amino Acids Signatures Associated with Clade B Transmitted/Founder and Recent Viruses"

_viruses, 2019, doi:10.3390/v11111012_

Round 1
Reviewer 1 Report
This manuscript describes a set of experiments and analysis, involving serum samples from HIV-1 infected individuals. The samples were used to reverse transcribe and amplify HIV-1 RNA envelope protein sequences, followed by analysis. Samples were selected to apparently reflect different timeline categories of infection. When amplified products were not obtained in sufficient numbers for the study, electronic sequences were curated from an electronic database.
The authors claim to have found “signature” amino acid sequences that reveal fundamental differences in virus phenotypes of different timeline categories of infection.
In the M&M section and throughout the manuscript, I found the paper would benefit from a clear table explaining/defining the different timeline categories of infection. This was confusing to me. Although the authors show a flow chart, the terms are not clear. For example, the authors use inconsistent nomenclature, or at least difficult for me to consistently follow, including all of these terms (and more) to describe the samples:
“Recent infection”
“Acute/TF”
“Early RC”
“Chronic”
“Transmitted”
“Founder”
“Recent”
“Acute/early”
“Early/Recent”
“actively HIV infected individuals”
“Acute HIV included TF virus (p24-antigen positive in the absence of HIV-1 antibody)”
“Recently HIV-1-infected (less than 136 days)”
“Chronic samples (WB+ and negative p24)” – line 83 p2
“Confirmed untreated clade B… (from LANL) as + ve control in addition to chronic virus envelope sequences” - p2 line 85 (which is it? Part of the data set for chronic, or control sequences?).
“Acute phase represents viral RNA, HIV p24 and negative antibodies” (repeat of earlier statements)
However, I think this problem can be easily fixed with a table and more consistent nomenclature and suggest a rewrite with this in mind.
More difficult to fix, is the surprising fact that only 2 “chronic” sequences were derived from their samples, excluding the curated sequences from a database.
In terms of M&M, no repeat of the RT PCR for the same samples from the same patient were done, to verify that the results are repeatable and not PCR errors.
The primers used are referred to as “appropriate”, but had no redundancy, so if the primers are for HIV, a subset of sequences would be expected to be amplified. Since HIV-1 exist as “quasispecies”, “eignefunction” or “viral swarms”, the study may have been biased towards some sequences.
In terms of the phylogenetic tree presented, I don’t think this is informative. It is not a consensus-bootstrapped tree, so the topology nodes are meaningless, even if the authors claim it to be the most frequent tree. Plus the font size is very small for a page, so becomes useless in printed versions of the manuscript.
Again, for the ridge plot patterns, it would be easier to follow if the nomenclature in the text and figures was simplified and identical.
If I understood correctly, most of the apparent genetic signature sequences identified, involve minor amino acid sequence frequency changes. For example, in figure 6, the GP41-CD-LLP-1 Domains sequence signature, is the presence of a minor R instead of a minor I; but the major number of both Chronic and TF sequences, both have an L at this position, number 959. Same for Figures 7 (at position 6, both sets of data show an I as the predominant sequence). Same for figures 8 and 9. Only in figure 10 and 11 is there a change in the predominant sequence at the positions indicated.
I could not find the apparently very significant R841I mentioned (p17, line 450 in the figures and presumed they meant the I959, in figure 6 (?). Unless the authors are referring to the very minor frequency R 841 in figure 11(?).
I am unable to judge the significance, or not, of the glycosylation site data.
The discussion should refer to figures when talking about/ interpreting the results. This would be very helpful.
Author Response
Response to Reviewer 1 Comments
For complete figure and table, please see the attached
Comments and Suggestions for Authors
This manuscript describes a set of experiments and analysis, involving serum samples from HIV-1 infected individuals. The samples were used to reverse transcribe and amplify HIV-1 RNA envelope protein sequences, followed by analysis. Samples were selected to apparently reflect different timeline categories of infection. When amplified products were not obtained in sufficient numbers for the study, electronic sequences were curated from an electronic database.
The authors claim to have found “signature” amino acid sequences that reveal fundamental differences in virus phenotypes of different timeline categories of infection.
In the M&M section and throughout the manuscript, I found the paper would benefit from a clear table explaining/defining the different timeline categories of infection. This was confusing to me.
Point 1: Although the authors show a flow chart, the terms are not clear.
Response 1: Thank you. We have corrected term in flow chart as follow in manuscript:
Figure 1. Flow Chart of HIV-1 positive samples analysed and the numbers of HIV-1 envelope sequences includes in the current study.
Point 2: For example, the authors use inconsistent nomenclature, or at least difficult for me to consistently follow, including all of these terms (and more) to describe the samples: “Recent infection”; “Acute/TF”; “Early RC”; “Chronic”; “Transmitted”; “Founder”; “Recent”; “Acute/early”; “Early/Recent”; “actively HIV infected individuals”; “Acute HIV included TF virus (p24-antigen positive in the absence of HIV-1 antibody)”; “Recently HIV-1-infected (less than 136 days)”
Response 2: Thank for the reviewer on this observation. We agree that the nomenclatures are inconsistent. We have now summarized them in point 5 of this response letter and inserted in manuscript as supplemental Table S1
Point 3: “Chronic samples (WB+ and negative p24)” – line 83 p2
Response 3: Thank you. We have rewritten the sentence of line 83 in manuscript.
Point 4: “Confirmed untreated clade B… (from LANL) as + ve control in addition to chronic virus envelope sequences” - p2 line 85 (which is it? Part of the data set for chronic, or control sequences?).
Response 4: Thank you for this observation. Yes, we considered these sequences as part of the data set and we have modified the text accordingly to make it clear. We have now corrected this sentence in the manuscript as follows: Confirmed untreated chronic clade B HIV-1 envelope sequences were also obtained from the Los Alamos sequence databases (LANL) as part of the data set for chronic infection in addition to chronic viruses’ envelope sequences obtained among public health laboratory of Quebec (LSPQ) serobank samples collection.
Point 5: “Acute phase represents viral RNA, HIV p24 and negative antibodies” (repeat of earlier statements).
Response 5: Thank you. We have moved the repeat statements in manuscript.
Point 6: However, I think this problem can be easily fixed with a table and more consistent nomenclature and suggest a rewrite with this in mind.
Response 6: Thank you for suggestion. We have summarized and corrected nomenclatures for samples and sequences description in manuscript and added a supplemental Table S1 as follow.
Table S1: Samples description: defining timeline categories of HIV-1 infection and referred nomenclatures.
Infection category |
Virus type |
Fiebig stage |
Duration |
Biomarkers |
Sample derived from |
Acute infection |
Transmitted/founder viruses (TF) |
Fieb. 1-2 |
14-21 days |
HIV-RNA+ and/or p24 Ag+, Western blot -, HIV antibody- |
Acutely infected individuals |
Early infection |
Recent viruses (RC) |
Fieb. 3-5 |
≤136 days according RITA testing (Sherir B. et al. 2016) |
HIV p24 Ag-, Western blot +, HIV antibody +, qualified as recent by RITA testing |
Recently infected individuals |
Chronic infection |
Chronic viruses (CH) |
Established |
>6 months Sherir B. al. 2016) |
HIV antibody + |
Chronically infected individuals |
Point 7: More difficult to fix, is the surprising fact that only 2 “chronic” sequences were derived from their samples, excluding the curated sequences from a database. In terms of M&M, no repeat of the RT PCR for the same samples from the same patient were done, to verify that the results are repeatable and not PCR errors. The primers used are referred to as “appropriate”, but had no redundancy, so if the primers are for HIV, a subset of sequences would be expected to be amplified. Since HIV-1 exist as “quasispecies”, “eignefunction” or “viral swarms”, the study may have been biased towards some sequences.
Response 7: We agree with the reviewer that the amplified regions may have been possibly biased toward some sequences. However, we confirm that we have repeated the RT-PCRs for samples from the same patients using alternate PCR enzymes to increase success rate sensitivity and have found similar results.
As reported in the manuscript, key factors may have possibly influenced the RT-PCR success rate including lower viral load for chronic samples and samples qualities per se. We have used the same PCR condition for all samples including primers and enzymes. The above primers were used in our labs for several studies and provided valid results with successful RT-PCR amplification (Asin-Milan O, et al. 2014.). These primers were also used previously in earlier studies for HIV-1 envelope amplification (Revilla A, et al. 2011; Shcherbakova NS et al., 2014, among others).
The chronic HIV-1 envelope sequences electronically selected in LANL were found to be valid and were already included in earlier published articles (Gnanakaran, S., et al. 2011). We thought that it would be useful to include them as part of this study. Similar to our approach, some of these last-mentioned studies also used only electronically sequences data in combination to in-house sequences and have provided results and publish their research results in literature (Gnanakaran, S., et al. 2011). As mentioned in manuscript, we acknowledged that the lower chronic HIV-1 envelope sequences from the same population constitutes a limitation to the current study but didn’t influenced results and research validity. To minimize the bias, sequences selected in LANL HIV sequences databases were limited to North American HIV+ individuals. Of note, both U.S. and Canadian HIV-infected populations presented the same risk of infection acquisition (MSM, IDU).
Point 8: In terms of the phylogenetic tree presented, I don’t think this is informative. It is not a consensus-bootstrapped tree, so the topology nodes are meaningless, even if the authors claim it to be the most frequent tree. Plus the font size is very small for a page, so becomes useless in printed versions of the manuscript.
Response 8: As suggested by the reviewers (Reviewer 1), we moved the phylogenetic tree from the manuscript. However, the small size was shown in figure insertion in word doc format if not, in TIFF format, it is better clear.
Point 9: Again, for the ridge plot patterns, it would be easier to follow if the nomenclature in the text and figures was simplified and identical.
Response 9: Thank you. We have simplified the nomenclature of ridge plot patterns in text and in figures now (see Figure 2-4).
Point 10: If I understood correctly, most of the apparent genetic signature sequences identified, involve minor amino acid sequence frequency changes. For example, in figure 6, the GP41-CD-LLP-1 Domains sequence signature, is the presence of a minor R instead of a minor I; but the major number of both Chronic and TF sequences, both have an L at this position, number 959. Same for Figures 7 (at position 6, both sets of data show that an I as the predominant sequence). Same for figures 8 and 9. Only in figure 10 and 11 is there a change in the predominant sequence at the positions indicated.
Response 10: Thank you for your observations. We agree with you that the genetic signature identified involve minor amino acids sequence frequency changes in the same HIV infection status virus’s envelope. This minor change was shown in the same category of infection, for example for TF viruses or chronic.
For the current study, the objective was to compare the amino acid changes across envelope sequences, position by position (from 1-856 of Env), and to compare frequencies between two timeline categories of infection (TF vs CH). Since these changes might be minor in the same category, we were interested to study its significance (difference) with other categories. This comparison may help identifying which mutation is significantly selected by TF or CH viruses, which might be considered as a genetic signature associated with other type of viruses.
The amino acids change frequency (major or minor) associated with TF viruses compared with chronic ones are reported in Table1 including results of statistical analysis.
Given the extremely high level of HIV-1 sequence polymorphism and diversity specifically in the envelope glycoprotein. We believe that any genetic changes either minor or discrete in this genomic region may impact viral fitness and its phenotypic property also observed in earlies study (Gnanakaran S., et al. 2011). Identifying such genetic traits or signatures may help better understanding genetic mechanism developed by TF viruses during viral transmission.
Point 11: I could not find the apparently very significant R841I mentioned (p17, line 450 in the figures and presumed they meant the I959, in figure 6 (?).
Response 11: Effectively the R841I signature was apparently minor in the TF virus envelope sequences. However, compared to chronic virus envelope sequences and based on statistical analysis results, the difference is highly significant between the two types of infections status. Therefore, we may consider this mutation as a characteristics of TF viruses. We think that, this observation in combination with other factors may influence viral fitness during HIV transmission.
Point 12: In line 450?
Response 11: We mean in line 450 that an isoleucine (I) amino acid signature was localized at position 959 of alignment position and referred to HXB2 numbering, it is at position 841 (R841I). We have corrected sentence in manuscript.
Point 13: Unless the authors are referring to the very minor frequency R 841 in figure 11(?).
Response 13: In figure 11 we wanted to highlight that the Valine (V) amino acid constitutes a genetic signature and was localized at position 876 of alignment and 751 referring to HXB2 numbering (D751V). Observation was based on the statistical analysis results summarized in Table 2.
Point 14: I am unable to judge the significance, or not, of the glycosylation site data.
Response 14: Thanks, you for observation: The summary results of statistical analysis (Logistic regression) were reported in supplemental Table S5 as follow.
Point 15: The discussion should refer to figures when talking about/ interpreting the results. This would be very helpful.
Response 15: Thank you for the suggestion. We have now modified the discussion section to refer to figures when appropriate.

Reviewer 2 Report
The Viruses manuscript #592615 by Kafando et al. presents a comparative analysis of HIV-1 Env sequences from viruses in different stages of disease progression. The specimens for acute and early stages of infection were sequenced by the authors, while the sequences for the chronic stage of infection were mostly from the LANL database (only 2 out of 105 sequences were from their sequencing reactions). The authors didn’t perform single genome sequencing, rather bulk NGS was performed on non-limiting dilution samples. The authors compare their consensus sequences thus generated with those from the LANL database and reach some conclusions. They propose that some of the differences observed between these two groups of sequences (specifically R841I, and K6I), play an important role in the establishment of chronic infection. Unfortunately, it is not clear that the differences observed are significant enough, to justify this conclusion. Among all the HIV proteins, Env is subjected to extensive selection pressures and bottlenecks, resulting in highly varied sequences even among viruses within a single individual. Differences in host genetic influences can have large impact on the evolution and diversification of HIV quasi species. The progression of Env sequence evolution would have been better addressed in a longitudinal study of Env sequences in patients. Some of the mutations found in other studies were not picked up here (discussion). Are the proposed Env sequence differences between Transmitted/founder (TF), recently infected and chronic (CH) virus samples of significance? This is not evident to the reviewer as discussed in points below. Several specific amino acid substitutions have been proposed to be a signature of either TF or CH virus. If bonafide, it should be demonstrable that using these signatures, it is possible to isolate/enrich for TF, RC and CH sequences from a sequence database containing all sequences.
Specific comments;
1) In the acutely infected patient samples, no information is provided about the drug resistance profile of the viruses, and the complexity of the viral populations? What information is known about the mode of HIV acquisition in these patients? These factors would influence the diversity of the viruses in the early stage samples.
2) The distributions shown for the number of N-glycosylation sites (NX[ST] pattern recognition using the LANL tools) in V3 loop are statistically different by using the Kruskal-Wallis test. The table S4 shows how close the numbers, median and range are. In discussion (line 426) “...the N-glycosylation sites of acute/TF viruses were significantly less glycosylated than chronic ones.” It is not clear how those authors get this information. The prediction algorithm only looks for the amino acid sequence pattern and does not predict the glycosylation outcome. What would be the outcome if other glycosylation site prediction algorithms were used?
3) The charge calculations performed by the LANL analysis tools – how is the charge calculated, and at what pH is the charge being reported? The Kruskal-Wallis test predicts that the “distributions’ for charge between Chronic (CH) and Transmitted/founder (TF) is statistically different, as also the differences between TF-RC-CH. The table S4 shows that the V3 net charge, median and range of the distributions are actually remarkably similar. In general, the comparative distributions seen in figures 3, 4 and 5 have similar shapes and overlay well on the horizontal scale, ie; they are not shifted in register. The main differences are predominantly in the fine structure of the distribution profiles, and this is borne out by the data in table S4.
4) R841I in the LLP-1 region of the cytoplasmic tail was enriched in TF viruses, and K6I in the signal peptide were found to be enriched in CH viruses. It is clear that this is only enrichment in the populations, and not an absolute requirement as seen in Fig 6 and 7. Unfortunately, because the analysis is using bulk sequencing, it is not possible to glean further information to identify any linkage to other changes in Env. In the Weblogo comparisons in Fig 8 -11 TF and RC sequences are combined to compare to CH. What was the rationale for this? What happens if they are treated separately as in fig 6 and 7, and how does the data in Fig 6, 7 change if the TF sequences are clubbed with RC? Comparing Fig 7 to Fig 9 – it is clear that M40I is picked up as statistically significant when TF and RC data are pooled before comparison (Fig9), but not when TF data alone is compared (Fig7) to CH.
5) The default Y-axis in Weblog is information bits; Figures 8-11 in this manuscript report amino acid frequencies; do the numerical values on the y-axis remain the same, or have they been normalized to the number 4?
6) Table 2 lists changes between CH and recent Env sequences; what are the changes F717F, T132T, R747R, R744R, T236T? These don’t look like changes.
7) Line 102; The amount of RNA that is used for reverse transcription should be mentioned, rather than the volume.
8) Line 77: first, not firstly.
9) General comments; the axis in figs 3-5 are not clear; I could not read the actual values. The labels HXB2 nt [aa] in figs 6-11 is confusing and should be elaborated. Table 2 can be reformatted are there is word wrap observable. Details of all the bioinformatics steps can be described better to allow for reproducibility.
Author Response
Response to Reviewer 2 Comments
For complete table, please see the attached
Comments and Suggestions for Authors
The Viruses manuscript #592615 by Kafando et al. presents a comparative analysis of HIV-1 Env sequences from viruses in different stages of disease progression. The specimens for acute and early stages of infection were sequenced by the authors, while the sequences for the chronic stage of infection were mostly from the LANL database (only 2 out of 105 sequences were from their sequencing reactions). The authors didn’t perform single genome sequencing, rather bulk NGS was performed on non-limiting dilution samples. The authors compare their consensus sequences thus generated with those from the LANL database and reach some conclusions. They propose that some of the differences observed between these two groups of sequences (specifically R841I, and K6I), play an important role in the establishment of chronic infection. Unfortunately, it is not clear that the differences observed are significant enough, to justify this conclusion. Among all the HIV proteins, Env is subjected to extensive selection pressures and bottlenecks, resulting in highly varied sequences even among viruses within a single individual. Differences in host genetic influences can have large impact on the evolution and diversification of HIV quasi species. The progression of Env sequence evolution would have been better addressed in a longitudinal study of Env sequences in patients. Some of the mutations found in other studies were not picked up here (discussion). Are the proposed Env sequence differences between Transmitted/founder (TF), recently infected and chronic (CH) virus samples of significance? This is not evident to the reviewer as discussed in points below. Several specific amino acid substitutions have been proposed to be a signature of either TF or CH virus. If bonafide, it should be demonstrable that using these signatures, it is possible to isolate/enrich for TF, RC and CH sequences from a sequence database containing all sequences.
Specific comments;
Point 1: In the acutely infected patient samples, no information is provided about the drug resistance profile of the viruses, and the complexity of the viral populations? What information is known about the mode of HIV acquisition in these patients? These factors would influence the diversity of the viruses in the early stage samples.
Response 1: We wanted to thank you for time taken and for better analyzing our manuscript. We agreed with you that acutely infected patients samples didn’t contains information’s about drug resistance profiles and the complexity of viral populations not mentioned. As that, all samples for acutely infected patients were selected from first time HIV diagnosis before enrollment in any antiretroviral therapy and based on P24 Ag+ before seroconversion absence of HIV antibody. At this state, we think that it is not necessary for use to provide drug resistance profile. In Canada, and for HIV infected populations include in this study, majority of them are Men who sex with men (MSM) and someone Injecting drug users (IDU) that were generally infected with clade B HIV-1.
We agreed with you that the route of HIV acquisition influence success of viral transmission to new recipient, the viral population (Quasispecies) and sequences diversity but not necessary the characteristics (polymorphism) the envelope sequences of the TF viruses themselves (Tully DC et al.. PLoS Pathog. 2016). We agreed with you that HIV-1 envelope Env is subjected to extensive selection pressures and bottlenecks at acute stage of infection. Unless, the viral population at acute state of HIV infection is known to be more homogenous than chronic as reported in the introduction of this study and publish elsewhere (Joseph SB et al. Nat Rev Microbiol. 2015, Kariuki SM, et al. Retrovirology. 2017; Parrish NF,. Proc Natl Acad Sci U S A. 2013, Ashokkumar M et al. J Virol. 2018; Parker ZF et al. J Virol. 2013; Keele BF et al. Proc Natl Acad Sci U S A. 2008)
For this study, we focused of characterizing and comparing genetic properties of HIV-1 envelope protein of CH, RC and TF viruses including loop lengths, number of n-glycosylation sites, sequences polymorphism and V3 net charge as publish by other authors (Gnanakaran S et al. PLoS Pathog. 2011; Chohan B. et al. J Virol. 2005…). To answer your question, we have mentioned the route of HIV acquisition for the study population in revised manuscript (section, description of specimens).
Point 2: The distributions shown for the number of N-glycosylation sites (NX[ST] pattern recognition using the LANL tools) in V3 loop are statistically different by using the Kruskal-Wallis test. The table S4 shows how close the numbers, median and range are. In discussion (line 426) “...the N-glycosylation sites of acute/TF viruses were significantly less glycosylated than chronic ones.” It is not clear how those authors get this information. The prediction algorithm only looks for the amino acid sequence pattern and does not predict the glycosylation outcome. What would be the outcome if other glycosylation site prediction algorithms were used?
Response 2: Thank you for your observations. We agreed with you but we also used regression logistic analysis to predict the number of glycosylation site outcome between TF and CH viruses envelope V3 as reported in Fig.2 and Table S4 (read Table S5). We have now completed table S5 in final manuscript including results of statistical analysis (median, range OR, P value). Results reported in discussion section was based on these statistical analysis (Table S5 see point 3). As shown, if numbers of observations were closely similar a little difference between two populations may be significant in statistical analysis generally. We have repeated analysis and have found same results using STATA software.
Point 3: The charge calculations performed by the LANL analysis tools – how is the charge calculated, and at what pH is the charge being reported? The Kruskal-Wallis test predicts that the “distributions’ for charge between Chronic (CH) and Transmitted/founder (TF) is statistically different, as also the differences between TF-RC-CH. The table S4 shows that the V3 net charge, median and range of the distributions are actually remarkably similar. In general, the comparative distributions seen in figures 3, 4 and 5 have similar shapes and overlay well on the horizontal scale, ie; they are not shifted in register. The main differences are predominantly in the fine structure of the distribution profiles, and this is borne out by the data in table S4.
Response 3: We tanks you for observation. The charge was automatically generated in LANL analysis tools. https://www.hiv.lanl.gov/content/sequence/VAR_REG_CHAR/index.html.
For Net charge calculations, we used default setting of net charge calculations that is computed with KRH = (+) and DE = (-). This observation was reported now in manuscript.
Conclusion of Env V3 net charge differences between CH and TF viruses sequences was based on logistic regression analysis that we haven’t reported in previous Table S4. As discussed in point 3, if numbers of observations were closely similar a little difference between two populations may be possibly significant in statistical analysis. Now, we have reported all statistical analysis in Table S5 (previous S4) including OR and P values as follow:
Point 4: R841I in the LLP-1 region of the cytoplasmic tail was enriched in TF viruses, and K6I in the signal peptide were found to be enriched in CH viruses. It is clear that this is only enrichment in the populations, and not an absolute requirement as seen in Fig 6 and 7. Unfortunately, because the analysis is using bulk sequencing, it is not possible to glean further information to identify any linkage to other changes in Env. In the Weblogo comparisons in Fig 8 -11 TF and RC sequences are combined to compare to CH. What was the rationale for this?
Response 4.1. Thanks for observations. We agreed with you that sequencing (NGS) may possibly influences analysis. But, for this study, all reads generate by sequencing were de novo assembled to generate consensus envelope sequences for each specimen. Consensus sequences of TF viruses were then aligned with HXB2 Env reference sequence, RC and CH ones at the same condition using MEGA7 software. Frequency of amino acid at each position of Env sequence (gp160) length from 1-856 were generated by Weblogo using (Data plaint text) output format. Distributions amino acids were analyzed statistically to identify any change and difference between TF, RC and CH viruses.
For this study, we first compared TF viruses sequences with RC and CH separately to identify any amino acids change developed very early as possible (Fiebig stage 1-2) with those developed at late stages of infection RC (Fieb. 3 to 5) and CH. The last ones may possibly constitute results of accumulate mutations or changes due to constant selection pressure on Env.
We then combined TF and RC sequences and compared with CH sequences as we wanted to know if considering all stages of recent infections sequences as used in earlies study (Gnanakaran et al. 2011), we may found same observations as previous ones. Many studies often consider early founder viruses as those infected in this window (Fieb 1-5), see Table S1 defining timeline categories of infection status and referred nomenclatures add. The current study aimed to identify genetic signatures following diseases progression beginning at different stages of early infection to chronic.
We think that identifying such early genetic traits as possible, may objectively reflect the characteristics of TF viruses. Since predominant of minor, these changes may constitute one part of mechanisms or factors helping success of HIV transmission and/or establishment of infection to new recipient in normal infection.
Point 4. What happens if they are treated separately as in fig 6 and 7, and how does the data in Fig 6, 7 change if the TF sequences are clubbed with RC? Comparing Fig 7 to Fig 9 – it is clear that M40I is picked up as statistically significant when TF and RC data are pooled before comparison (Fig9), but not when TF data alone is compared (Fig7) to CH.
Response 4.2. For this study, we wanted to identity the genetic signatures developed very early as possible after infection (acute stage) Table S1 before any accumulation of mutations or changes that may developed during diseases progression.
Deciphering the very early genetic events may help understanding genetic properties of TF viruses and factors that may influence HIV transmission success to new recipient. These Env. genetic information’s validate in phenotypic studies may help designing HIV preventive vaccine we think.
Point 5: The default Y-axis in Weblog is information bits; Figures 8-11 in this manuscript report amino acid frequencies; do the numerical values on the y-axis remain the same, or have they been normalized to the number 4?
Response 5. Thank for observation. Yes, default information’s in Weblogo Y-axis is normally sets in bits, probability, nats, kT, kJ/mol, kcal/mol. In this case they are normalized to the number 0 to 4.
Point 6: Table 2 lists changes between CH and recent Env sequences; what are the changes F717F, T132T, R747R, R744R, T236T? These don’t look like changes.
Response 6: Yes, you have reason. In these positions they are not amino acids change. But, when comparing CH and TF virus envelope amino acids frequencies in these positions, the differences were statistically significant. It mean that some amino acid was highly selected for CH or TF viruses. The difference may possibly due to others minors or discrete amino acids mutations developed elsewhere of one or other. For this study, discussion we focused on differences of amino acids change (substitution) between CH and TF by positions of their Env sequences. But, we have reported all significant differences identified in Table 2 for observation.
Point 7: Line 102; the amount of RNA that is used for reverse transcription should be mentioned, rather than the volume.
Response 7: Extracted RNA (approximately 60–80 µl) was immediately reverse transcribed or stored at −80°C for reference use. We agreed with you that concentration of extracted RNA should be mentioned. However, at this state (RNA extraction) we have not measured the RNA amount. All extracted suspension should be contained HIV RNA but, after reverse transcription all them are not successfully amplified as reported in Figure 1 of manuscript. We have measured amount of amplicon after second PCR (Nested PCR) at concentration of purified DNA of 0.2 ng/µl for sequencing.
Point 8: Line 77: first, not firstly.
Response 8: Okay, thanks. We have corrected this word in manuscript as suggested.
Point 9: General comments; the axis in figs 3-5 are not clear; I could not read the actual values. The labels HXB2 nt [aa] in figs 6-11 is confusing and should be elaborated. Table 2 can be reformatted are there is word wrap observable. Details of all the bioinformatics steps can be described better to allow for reproducibility.
Response 9: Thanks for general comments. Figs 3-5 read now 2-4 are corrected and modified. Figs 6-11 (read now 5-10) were elaborated. Table2 is reformatted and bioinformatics sections are revised in manuscript.

Reviewer 3 Report
This manuscript presents a bioinformatic analysis comparing the Env sequences of transmitted/founder, recently infected, and chronic HIV-1 genomes. The authors report some statistically significant differences in lengths and charges in the variable loops of gp120. They also note some statistically significant sequence differences.
While I cannot criticize the analysis or the conclusions, the manuscript appears to be carelessly prepared and I found it extremely difficult to follow. There are many mistakes in the text. Here are a few examples: line 48: “The envelope glycoprotein…is the first component of viral genome…” (the glycoprotein is not a component of the genome, but is encoded by it); line 66: “It is important to study TF viruses that envelope protein…” (I believe a word is missing from this sentence); I could find no information about the primers SG3-lo and SG3-up; line 108 “which included 02 ul Superscript…” (should this be 0.2?)(and similarly in line 109); the polymerase is not identified (line 113) and the description of the PCR conditions (lines 114-117) is not clear; Figure 2 is never mentioned in the text; Figures 3-5 are extremely difficult to understand. The labeling of the vertical and horizontal axes is too small to read. Individual panels should be labeled in the figure itself—for example, V1 for Fig. 3a, TF, RC, and CH for the 3 graphs within Fig. 3a, etc. This information is in the manuscript but it should be in the figure, not somewhere else in the manuscript. It seems grandiose to state (line 404) “The main objective of the study was to determine the molecular characteristics of the variable loop”: there is really no discussion of molecular characteristics in the manuscript. I do not understand lines 479-480, “K6I may be considered as supplement selected amino acids necessary during disease progression in the chronic stage” . References 17 and 32 are identical, as are refs. 11 and 51.
Author Response
Response to Reviewer 3 Comments
For complete figure, please see the attached
Comments and Suggestions for Authors
This manuscript presents a bioinformatic analysis comparing the Env sequences of transmitted/founder, recently infected, and chronic HIV-1 genomes. The authors report some statistically significant differences in lengths and charges in the variable loops of gp120. They also note some statistically significant sequence differences. While I cannot criticize the analysis or the conclusions, the manuscript appears to be carelessly prepared and I found it extremely difficult to follow. There are many mistakes in the text. Here are a few examples:
Point 1: Line 48: “The envelope glycoprotein…is the first component of viral genome…” (the glycoprotein is not a component of the genome, but is encoded by it);
Response 1: Thank you for correction. We have modified the sentence in the manuscript as suggested.
Point 2: Line 66: “It is important to study TF viruses that envelope protein…” (I believe a word is missing from this sentence);
Response 2: Thank you for your observation. We have modified the text as follows in the manuscript: The envelope glycoprotein of HIV-1 is encoded by viral genome and this protein mediates the first contact with host cells. Even discrete or predominant, an amino acid change (mutations, insertions, and deletions) in this viral specific region merit further analysis. Such changes may be constituting an important genetic signature developed at the acute stage of infection and selected by TF viruses. They may constitute keys factors that influence viral fitness and enhancing HIV transmission to new recipient.
Point 3: I could find no information about the primers SG3-lo and SG3-up;
Response 3: Information’s for SG3-lo and SG3-up primers were referred to following articles and are now integrated to manuscript.
Revilla A, Delgado E, Christian EC, et al. Construction and phenotypic characterization of HIV type 1 functional envelope clones of subtypes G and F. AIDS Res Hum Retroviruses 2011; 27(8): 889-901. Shcherbakova NS, Shalamova LA, Delgado E, et al. Short communication: Molecular epidemiology, phylogeny, and phylodynamics of CRF63_02A1, a recently originated HIV-1 circulating recombinant form spreading in Siberia. AIDS Res Hum Retroviruses. 2014;30(9):912–919. doi:10.1089/AID.2014.0075 Asin-Milan O, Wei Y, Sylla M, Vaisheva F, Chamberland A, Tremblay CL. Performance of a clonal-based HIV-1 tropism phenotypic assay. J Virol Methods. 2014 Aug; 204:53-61. doi: 10.1016/j.jviromet.2014.04.004. Epub 2014 Apr 13. PubMedPMID: 24731927 Protocol for construction of hiv-1 functional envelope clones from plasma:http://www.euripred.eu/fileadmin/user_upload/Documenten/Information_and_products/SOPs/Protocol_HIV-1_functional_envelope_construction.pdf
Point 4: Line 108 “which included 02 ul Superscript…” (should this be 0.2?) (and similarly, in line 109);
Response 4: Thank you. No, it is 2 µl in our RT PCR protocol, we corrected this in the text as 2.0 µl
Point 5: The polymerase is not identified (line 113)
Response 5: Thanks for the observations. We have now identified the polymerase in the manuscript: It is the: Platinium® Taq DNA polymerase.
Point 6: and the description of the PCR conditions (lines 114-117) is not clear;
Response 6: We have now revised the PCR conditions section in manuscript.
Point 7: Figure 2 is never mentioned in the text;
Response 7: We have now removed Figure 2 from the manuscript as other reviewers (reviewer 1) did not find it informative for this paper. If not, in TIFF format, it is better clear compared to in-text word doc format. Phylogenetic also tree included most envelope sequences that may have reducing its font size and quality (N=231).
Point 8: Figures 3-5 are extremely difficult to understand. The labeling of the vertical and horizontal axes is too small to read. Individual panels should be labeled in the figure itself—for example, V1 for Fig. 3a, TF, RC, and CH for the 3 graphs within Fig. 3a, etc. This information is in the manuscript, but it should be in the figure, not somewhere else in the manuscript.
Response 8: We thank the reviewer for this comment. We agree and we have now corrected figures of concern (Figures 3-5). Please note that these figures are now referred to as Figures 2-4 since we excluded original Figure 2 from the corrected manuscript.
Figure 2. Number of N- glycosylation’s sites of HIV-1 Env Variable regions
Figure 3. Number of Env Variable regions loop lengths
Figure 4. HIV-1 Env loop 3 net charge
Point 9: It seems grandiose to state (line 404) “The main objective of the study was to determine the molecular characteristics of the variable loop”: there is really no discussion of molecular characteristics in the manuscript.
Response 9: We agree with the reviewer that we did not highlight this in the discussion section. Our study is focusing on the HIV-1 envelope variable characteristics in term numbers of N- glycosylation sites, loop lengths and V3 net charges as characteristics of TF viruses in earlier studies. This study aimed to verify these findings in the LSPQ serobank of TF HIV-1 envelope sequences. We have now modified the discussion section accordingly, Page 16, lines 417-441.
Point 10: I do not understand lines 479-480, “K6I may be considered as supplement selected amino acids necessary during disease progression in the chronic stage”.
Response 10: In lines 479-780, we mean that in the current study that chronic viruses selected Isoleucine (I) amino acid in position 6 of signal peptide compared Gnanakaran et al. 2011 study that have identified a histidine (H) in position 12 of signal peptide for chronic viruses. Yes, we agreed with you that this sentence is not clear. We have rewritten it accordingly in manuscript.
Point 11: References 17 and 32 are identical, as are refs. 11 and 51.
Response 11: Thanks for the observation. We have now removed the redundant references and corrected them in the manuscript (Ref 32 was moved and replaced by 17 and similarly 51 was replaced by 11).

Round 2
Reviewer 2 Report
The changes to the manuscript are mostly cursory and textual, though addition of some statistical parameters is welcome. Unfortunately, at least to this reviewer, the authors have not tried to address the technical deficiencies in their data analysis and interpretation. The authors agree with the lack of information which can confound analysis, such as drug resistance profiles of the TF viruses. The authors now use logistic regression analysis to claim statistical significance in differences in V5 loop length between TF, RC, and CH viruses. This analysis is also used the same for V3 charge. The authors should describe details of the logistic regression analysis now utilized. A reader should be able to redo all these analysis for their own comparisons and research. The question of how the Weblogo analysis, or indeed any of the other analysis, would have been impacted if different groupings were considered was not addressed. The Y Axis for all the WebLogo figures should read Normalized AA frequency, or the numerical value makes no sense. Nevertheless, the reviewer feels that the authors over interpret the statistical significance differences between the distribution profiles. As I have mentioned in the original comments, if these signatures are indeed representative of specific stages, it should be possible to test these conclusions. The authors should have tempered their conclusions. I also notice several typo mistakes which are now in the revised manuscript.
Reviewer 3 Report
The authors have corrected many of the technical flaws in the original version of the manuscript. However, the labeling in the ridge plots (now Fig. 2 and 3) is still too small to read. The changes in numbering of figures are incomplete, and were missed in lines 261-262; 308, and 353. What is “HIR-KE” in Table 1?
The English in the text is full of errors and needs to be corrected by a native English speaker.